# Safety and efficacy of an artificial intelligence-enabled decision tool for treatment decisions in neovascular age-related macular degeneration and an exploration of clinical pathway integration and implementation: protocol for a multi-methods validation study

Henry David Jeffry Hogg [1,2], Katie Brittain,[1] Dawn Teare,[1] James Talks,[2] Konstantinos Balaskas,[3,4] Pearse Keane,[3,4] Gregory Maniatopoulos[1,5]

For numbered affiliations see end of article.

**Correspondence to**
Dr Henry David Jeffry Hogg;
jeffry.hogg@ncl.ac.uk

## ABSTRACT

**Introduction** Neovascular age-related macular degeneration (nAMD) management is one of the largest single-disease contributors to hospital outpatient appointments. Partial automation of nAMD treatment decisions could reduce demands on clinician time. Established artificial intelligence (AI)-enabled retinal imaging analysis tools, could be applied to this use-case, but are not yet validated for it. A primary qualitative investigation of stakeholder perceptions of such an AI-enabled decision tool is also absent. This multi-methods study aims to establish the safety and efficacy of an AI-enabled decision tool for nAMD treatment decisions and understand where on the clinical pathway it could sit and what factors are likely to influence its implementation.

**Methods and analysis** Single-centre retrospective imaging and clinical data will be collected from nAMD clinic visits at a National Health Service (NHS) teaching hospital ophthalmology service, including judgements of nAMD disease stability or activity made in real-world consultant-led-care. Dataset size will be set by a power calculation using the first 127 randomly sampled eligible clinic visits. An AI-enabled retinal segmentation tool and a rule-based decision tree will independently analyse imaging data to report nAMD stability or activity for each of these clinic visits. Independently, an external reading centre will receive both clinical and imaging data to generate an enhanced reference standard for each clinic visit. The non-inferiority of the relative negative predictive value of AI-enabled reports on disease activity relative to consultant-led-care judgements will then be tested. In parallel, approximately 40 semi-structured interviews will be conducted with key nAMD service stakeholders, including patients. Transcripts will be coded using a theoretical framework and thematic analysis will follow.

## STRENGTHS AND LIMITATIONS OF THIS STUDY

⇒ This non-inferiority study examines an artificial intelligence (AI) use case supported by patients and the public, with scope for improving the efficiency with which limited human resources are applied across the National Health Service and elsewhere.

⇒ This AI use case has a high degree of explainability as it relies on reviewable tissue segmentation and a simple rule-based decision tree which mimic widely used treatment paradigms.

⇒ The multi-methods approach allows insights beyond efficacy alone to consider the effectiveness, mechanisms and system impacts of this complex intervention.

⇒ Due to the preclinical nature of this study, stakeholder perspectives concerning this AI-use case will be hypothetical rather than from direct experience.

⇒ As the non-inferiority study uses retrospective data only, the results will not reflect the challenges of real-time AI deployment which will require onward study.

**Ethics and dissemination** NHS Research Ethics Committee and UK Health Research Authority approvals are in place (21/NW/0138). Informed consent is planned for interview participants only. Written and oral dissemination is planned to public, clinical, academic and commercial stakeholders.

## INTRODUCTION

A threat to the efficacy of UK National Health Service (NHS) ophthalmology services is posed by insufficient ophthalmologist availability, in the face of a growing clinical

need.[1 2] Ophthalmology continued to incur more clinic appointments than any other NHS specialty through the COVID-19 pandemic, with age-related macular degeneration (AMD) as the most common cause for these appointments.[2] AMD is the leading cause for certification of visual impairment in the UK and affects just over a quarter of Europeans aged over 60 to some extent.[3 4] Only the disease course of a severe form of AMD, complicated by choroidal neovascularisation (nAMD), can benefit from current treatment and is thought to affect 1.4% of Europeans aged over 60.[3] Real-world data from an NHS centre showed that even before the impact of the COVID-19 pandemic, delay in delivering planned treatments for nAMD was common and associated with sight loss for the patients concerned.[5] This same NHS centre had recently displayed better nAMD visual outcomes than a mean of 12 large UK centres, suggesting the observed treatment delays are likely to be representative of the whole nation.[6] Since these observations, the COVID-19 pandemic has introduced additional nAMD treatment delays and avoidable sight loss, further developing the urgent need to augment clinical capacity for nAMD treatment.[7]

Clinical artificial intelligence (AI) is an increasing target of investment from government, industry and academia, often with the hope of increasing service capacity independent of the limits posed by clinician availability.[8–10] Ophthalmology is one of the most heavily researched specialties in AI medical image analysis with the non-contact infra-red based imaging modality, optical coherence tomography (OCT), accounting for much of this.[11] Retinal OCT scans form the substrate for several clinical AI tools, which anatomically segment the images to support various clinical decisions including triage and diagnosis.[12] The impact of these clinical AI functions on the efficacy and efficiency of primary to secondary care referrals in ophthalmology are the subject of ongoing prospective clinical studies.[13 14] However, an even greater cumulative demand on ophthalmologists' time is making decisions about the treatment of nAMD within secondary care. Treatment takes the form of intravitreal injections (IVIs) of anti-vascular endothelial growth factor (anti-VEGF) agents which are recommended for variable periods of time and at varying intervals depending on changes in the retina observed through OCT.[15 16] While the decisions about treatments often involve ophthalmologists, the injections themselves are increasingly performed independently by specialised nurse practitioners and allied health professionals.[17 18] Given the highly protocolised, OCT-dependent, nAMD treatment pathway and pre-existent clinical AI capable of independently quantifying the OCT features which inform that treatment, there is a clear opportunity for AI-enabled nAMD services to expand capacity and reduce avoidable real-world sight loss.[12 16]

Despite this tension for change, clinical AI is yet to be implemented within routine NHS ophthalmic practice. This is at least partly due to the lack of evidence to forecast the effectiveness of clinical AI in the real-world setting

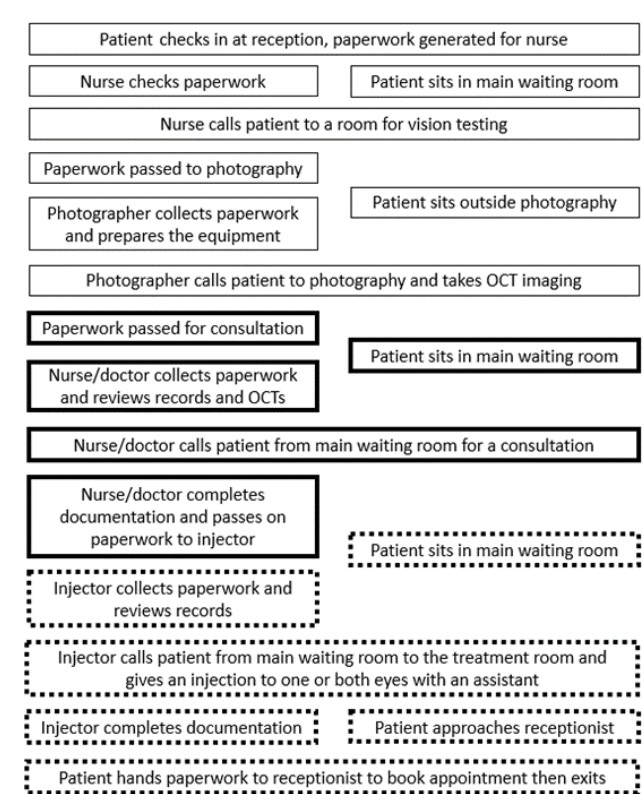

**Figure 1** Process map of current 'one-stop' macular clinic visits for a patient. Some steps have the potential to be substituted by AI-enabled treatment decisions based on OCT imaging (thick black line), some could be relocated to primary care optometry (thin black line) and some could also be decentralised using community outreach treatment services (dashed black line). AI, artificial intelligence; OCT, optical coherence tomography.

and how it will interact with the various stakeholders and contexts which must host the technology.[19] Alongside other innovations such as portable OCT technology and community outreach treatment services, AI may enable even greater changes to the way in which macular services are delivered (figure 1).[20 21] The complexity derived from these interdependent technologies, contexts and stakeholders must be better understood if the implementation is to optimise patient benefit from AI-enabled nAMD services. Given the years for which quantitative evidence of efficacy has existed for many clinical AI tools without subsequent real-world adoption, it may be that qualitative explorations of the complex implementation context are in fact the more urgent requirement.[11 22 23]

This single-centre multi-methods study aims to generate the necessary evidence by external validation of the technology with real-world data and exploring the interdependent factors which will influence the implementation of AI-enabled nAMD services. In doing so, we aim to move beyond efficacy to consider intervention effectiveness and construct a theoretical basis to guide future implementation strategies.[22]

A quantitative study will be performed to test the non-inferiority of AI-enabled reports of nAMD disease activity against judgements made in consultant-led-care.

> **Box 1    Summary of neovascular age-related macular degeneration (nAMD) treatment at Newcastle upon Tyne Hospitals National Health Service Foundation Trust**
>
> **Loading protocol:** Starts at diagnosis of nAMD and consists of three anti-VEGF IVIs at 4-week intervals (visits 2 and 3 do not include consultation or imaging), followed by IVIs at 8-week intervals, or less if signs of disease activity persist, for the remainder of the year.
>
> **Treat-and-extend protocol:** Starts one year after treatment initiation and dictates that the interval of anti-VEGF IVIs is increased in 2-week increments until evidence of disease activity is noted at which point the treatment interval is reduced. If extension beyond a certain interval is noted to result in observable disease activity a (unspecified) number of times, then that interval ceases to be modified.
>
> **Pro-re-nata protocol:** Initiated as a joint decision for patients who appear to have little or no disease activity having been on one of the other two protocols. Here, it is not assumed that an IVI will be given at each review, but only if evidence of disease activity is noted. The observation of returning disease activity may also lead to the return to one of the other two protocols.
>
> IVI, intravitreal injection; VEGF, vascular endothelial growth factor.

> **Box 2    Eligibility criteria for patients and clinic visits**
>
> Inclusion criteria
> ⇒ **Eye diagnosed with nAMD.**
> ⇒ **One or more prior anti-VEGF IVI at NuTH.**
> ⇒ Co-located 25 slice, fovea centred OCT imaging available for both the included and prior visits.
> ⇒ Clinic visit note states intended IVI interval.
> ⇒ Clinic visit included same-day IVI.
> Exclusion criteria
> ⇒ **Retinal diagnosis other than nAMD in included eye.**
> ⇒ Visit before 2016.
> ⇒ Visit during or after March 2020.
> ⇒ Visits conducted under the pro-re-nata treatment protocol.
>
> Eligibility criteria for patients is represented in bold.
> IVI, intravitreal injection; nAMD, neovascular age-related macular degeneration; NuTH, Newcastle upon Tyne Hospitals; OCT, optical coherence tomography.

In addition, a qualitative study based on semi-structured interviews with key stakeholders and patients will be conducted to explore where the clinical AI could sit within the nAMD care pathway and the factors likely to influence its implementation.

## METHODS AND ANALYSIS
### Quantitative methods
#### Sampling method
The electronic medical record (EMR) at Newcastle upon Tyne Hospitals NHS Foundation Trust (NuTH) will be searched to identify clinic visits where individuals received anti-VEGF IVI to treat nAMD. Patients at NuTH are treated under three different regimens dependent on the length of their diagnosis and joint decisions between clinicians and patients: loading, treat-and-extend and pro-re-nata (PRN) (box 1). The treatment intervals most commonly vary between 4 and 16 weeks. During the period through which data will be collected aflibercept and ranibizumab were the anti-VEGF treatments in use for nAMD.

Eligible visits from this dataset of 70 884 will be randomly selected for manual review of patient EMR files to screen against eligibility criteria (box 2). These criteria exclude clinical visits that took place before current treatment and OCT imaging protocols were established or after treatment decisions may have been influenced by the COVID-19 pandemic. They also exclude visits conducted under the PRN treatment protocol as all visits on this protocol meeting the inclusion criterion for a same day anti-VEGF IVI must have been judged by the clinician to show disease activity and will therefore not be representative of PRN clinic visits generally. To maximise their relevance to the research question alongside feasibility and rigour within a complex real-world dataset, the eligibility criteria and systematic screening approach through which they will be applied were iteratively designed and trialled by authors with clinical, operational and statistical expertise inside and outside of NuTH (HDJH, SJT, PK, KBalaskas and DT) (online supplemental file 1). Having reached a consensus on the eligibility criteria and the screening approach, a single researcher with 9 years of clinical experience at NuTH will perform data collection (HDJH), to support the consistency of their application and ensure fluency with local clinical and digital practices.

### Data collection and processing
For each included clinic visit the following data will be recorded to characterise the dataset:
► Anonym for the individual
► Eye laterality
► Gender
► Self-reported ethnicity
► Home address postal code stem
► Individual's age at that visit.

The following will be recorded to send anonymised to Moorfields Reading Centre to generate a report of disease stability or activity to act as an enhanced reference standard for each visit. This information will also facilitate more meaningful post-hoc error analysis to explore the mechanisms of failure which the AI-enabled tool may exhibit. The findings from these analyses are a secondary outcome of the study and will help to delineate any groups of cases for which the tool's performance needs to be monitored and improved in further work, or for which only clinician judgements should be applied. This list was developed through additions to a proforma from a recent exemplar protocol[24]:
► OCT and visual acuity (VA) for that visit and the prior.
► VA for the fellow eye at that visit.
► Time since first nAMD treatment at that visit.
► Total IVIs for nAMD in that eye up until that visit.
► Observed interval since that eye's last IVI.
► Presence or absence of evolving macular haemorrhage being recorded.

► Time since increasing disease activity was last observed.

► Treatment interval associated with that observation.

The following data will also be recorded from each consultant-led-care visit to assess the primary and secondary outcomes and will not be sent to Moorfields Reading Centre:

► Judgement of disease activity or stability.

► Planned interval to next IVI.

► Professional group conducting the consultation.

► Treatment protocol the visit was conducted under (box 1).

Separately, the present and prior pairs of OCT images relating to the same clinic visits will be transferred to Moorfields Eye Hospital NHS Foundation Trust for AI-enabled retinal segmentation.[12] The differences in retinal tissue volumes will be used in a rule-based decision tree to produce an AI-enabled binary report of disease activity or stability for each included visit.

### AI-enabled decision tool

The intervention to be tested on retrospective OCT imaging data is a deep learning tool with a U-net architecture with previously published details of training and validation.[12] It can produce volume quantification for the neurosensory retina, retinal pigment epithelium, fibrovascular pigment epithelium detachment, drusenoid pigment epithelium detachment, subretinal hyper-reflective material, subretinal fluid, intraretinal fluid, posterior hyaloid, epiretinal membrane and serous pigment epithelium detachment. An initial decision tree will report disease activity when the volumes of neurosensory retina, subretinal fluid or intraretinal fluid increase >5% between the current and prior visit OCTs.[25] This decision logic is based on a recent consensus from UK medical retina experts on treat-and-extend protocols for nAMD.[16] The exact tissue group contributors and decision thresholds for inter-visit changes in each of these tissue groups will be iterated on using an embedded pilot dataset described further below. This binary output was preferred over a scalar recommendation of treatment interval to preserve the tools' value across different treatment protocols (box 1). The choice of a binary output will also help the generalisability of results, as the translation of OCT findings into treatment decisions varies internationally.

### Outcomes measures

Over-treatment marginally increases the cumulative risk of IVI complications and the cost to the provider, but justifiably the main concern of patients and carers consulted in designing this study was sight loss through under-treatment.[5] Consequently, the probability of AI-enabled reports of disease stability being correct relative to judgements made in real-world consultant-led care has been taken as the most clinically relevant measure of diagnostic accuracy. This has led to a non-inferiority design with the relative negative predictive value (NPV) as the primary outcome.

Secondary outcomes will be:

► A comparison of other standard diagnostic accuracy metrics between AI-enabled reports of disease activity and judgements from consultant-led-care accompanied by confusion matrices (table 1).

► A comparison of the diagnostic accuracy of the five healthcare professional groups conducting consultations in consultant-led-care (nurses, optometrists, ophthalmology specialty trainees, medical retina sub-specialty fellows, medical retina sub-specialty consultants).

► A comparison of the treatment intervals recommended in real-world consultant-led care and the treatment intervals that would be derived from AI-enabled reports of disease activity given the treatment protocol.

► A case-by-case exploration of false-positive and false-negative reports of disease activity from the AI-enabled decision tool and consultant-led care.

### Justification of study design and sample size

An enhanced reference standard is required to facilitate comparison between a potential AI-enabled nAMD service and the real-world gold standard, consultant-led-care. Moorfields Reading Centre has an international reputation and track record in meeting this need for prior studies and will receive imaging and clinical data for each case in the present study to generate an enhanced reference standard.[12 13 24] The binary decision under examination for each included visit is whether the data suggest disease stability or activity. This simplification of the scalar number of weeks between treatments or the three-option decision regarding treatment interval maintenance,

**Table 1** Template confusion matrix showing the different possible classification of artificial intelligence (AI)-enabled reports of disease activity and judgements from consultant-led-care (CLC) for each eligible case

| | Moorfields reading centre identifies disease stability (negative) | Moorfields reading centre identifies disease activity (positive) |
|---|---|---|
| AI identifies disease stability (negative) | AI True negative | AI False negative |
| AI identifies disease activity (positive) | AI False positive | AI True positive |
| CLC identifies disease stability (negative) | CLC True negative | CLC False negative |
| CLC identifies disease activity (positive) | CLC False positive | CLC True positive |

extension or reduction is more broadly relevant to real-world practice where clinician and patient preferences about how treatment intervals should be altered on account of disease activity vary (box 1).[16] It also lowers the risk of inappropriately labelling a decision made in consultant-led-care with subtle influences from patient or clinician preference as 'incorrect'. Whether binary judgements of disease activity (positives) or stability (negatives) from consultant-led-care and the AI-enabled decision tool are labelled as true or false will be decided by the independent judgement of the Moorfields Reading Centre (table 1).

Treatment decision data to support power calculations are unavailable for this novel use case. Considering a different use case, the deep learning model to be applied within the proposed AI-enabled decision tool has demonstrated equivalent or superior retinal diagnostic performance to consultant specialists. Relative to final real-world clinical diagnoses it produced an area under the receiver operating characteristic curve of 0.99 for retinal diseases including nAMD.[12] This performance was dependent on the same intermediate anatomical segmentation step that will form the basis for the AI-enabled decision tool for nAMD treatment proposed here. This has supported the feasibility of the current proposal but does not provide the level of certainty required to perform a robust power calculation, which requires sufficiently accurate estimates of the NPV of both judgements from consultant-led-care and AI-enabled reports from paired data.[26] Therefore, a pilot dataset will be collected and sent for independent processing by Moorfields Reading Centre and the AI-enabled decision tool to supply these estimates. Prior work has established that for binary outcomes such as the one under study, little improvement is seen in estimating

precision or bias by increasing the size of the pilot dataset above 100.[27] A preliminary review of 100 NuTH consultant-led-care nAMD clinic visits under loading or TEX treatment protocols found that 79 reported disease stability, and would be classed as negative cases which could contribute to estimating NPV. Given that about 79% of eligible visits are classed as negative the randomly sampled pilot dataset required to accrue 100 negative cases is expected to be around 127 cases (100/0.79), but the consultant-led-care judgement for each case will be reviewed as it is curated to ensure the pilot dataset contains 100 negative cases. The estimated NPV of AI-enabled reports and judgements from consultant-led-care will be derived from this pilot dataset and inform a power calculation. From this, the number of additional eligible cases required to be collected and processed by Moorfields Reading Centre and the AI-enabled decision tool to test the non-inferiority of AI-enabled report NPV relative to judgements from consultant-led-care will be established.[26] This power calculation will include a significance level α=0.05, a power β=0.90 and a relative non-inferiority margin of δ=0.90. Although this non-inferiority margin is relative, it will be similar and no larger than an absolute equivalent. This allowed the application of 10% non-inferiority margins applied in comparable studies.[24 28] In evaluating this non-inferiority margin it is helpful to remember that the least desirable outcome of a false negative would yield a 2 or 4 weeks delay for the next planned treatment rather than treatment cessation and that 22% of patients are estimated to experience more than 4 weeks of delay to treatment in a year in current consultant-led care.[5] As such the null hypothesis will be rejected if the lower confidence limit for the relative NPV of AI-enabled reports is greater than 0.90.[26] This will be

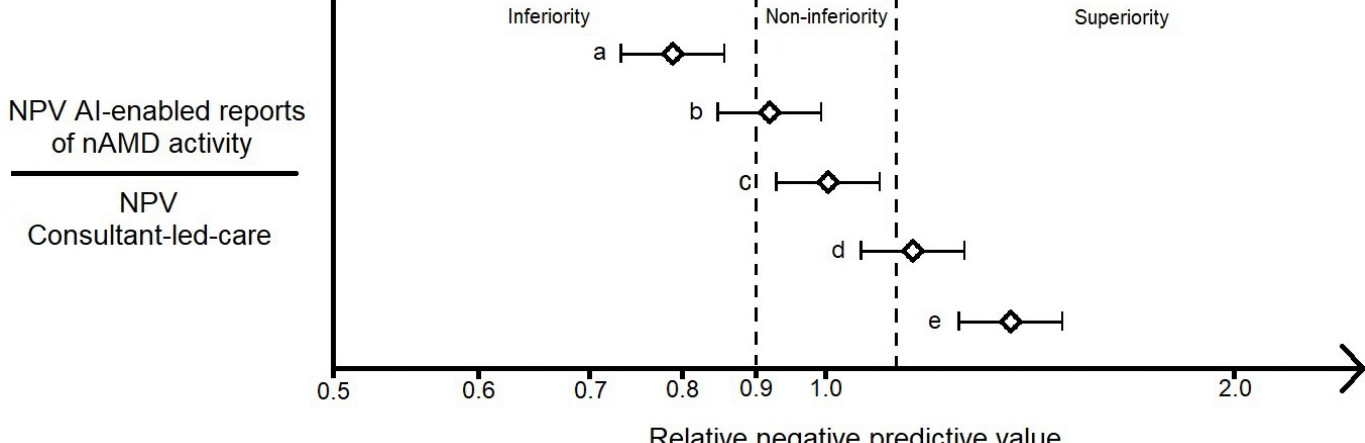

**Figure 2** Forest plot template for the relative negative predictive value (NPV) of artificial intelligence (AI)-enabled reports of neovascular age-related macular degeneration (nAMD) disease activity versus judgements from consultant-led-care relative to an enhanced reference standard from Moorfields Reading Centre. The non-inferiority and superiority margins are marked by dashed vertical lines on a logarithmic scale at 0.90 and 1.11, respectively. Potential outcomes for the non-inferiority test include scenario: (A) AI-enabled reports are inferior to judgements from consultant-led-care; (B) non-inferiority of AI-enabled reports to judgements from consultant-led-care is not demonstrated; (C) AI-enabled reports are non-inferior to judgements from consultant-led-care; (D) AI-enabled reports are non-inferior to judgements from consultant-led-care but not superior; (E) AI-enabled reports are superior to judgements from consultant-led-care.

visually presented with two-sided 95% CI to explore the possibility of superiority with a threshold γ of 1/ δ; γ=1.11 (figure 2). While the dataset will offer other important exploratory insights, the potential impact of the primary outcome on the translation of this AI-enabled decision tool and threat to the study's feasibility from ambitions outside this scope, has prevented any plans to proactively power the sample size for secondary outcomes.

### Data analysis

The relative NPV of AI-enabled reports/judgements from consultant-led-care will be calculated with 95% CIs to see if the inferiority, non-inferiority or superiority of AI-enabled reports of disease activity can be established (table 1). For secondary outcomes, diagnostic accuracy statistics for each group will be reported descriptively with 95% CIs for each group, along with confusion matrices. Clinical and imaging data from cases of false positives and false negatives of the AI-enabled reports will be reviewed by clinical members of the team, supported by the AI development team where necessary, to try to understand the mechanisms and potential consequences of AI-enabled decision tool failures. Standards for reporting diagnostic accuracy studies (STARD) will be used alongside the STARD-AI extension if available at the time of writing.[29 30]

### Qualitative methods

#### Sampling

A recent systematic review outlined the importance of all stakeholder perspectives in understanding the interdependent factors that influence clinical AI implementation.[31 32] Despite this, the qualitative literature regarding clinical AI is dominated by healthcare professionals' perspectives and has particularly limited representation from carers.[31] Consequently, the present study will aim to explore perspectives from all stakeholders. This will include patients from the NuTH nAMD clinic, carers of patients from the same clinic, doctors, nurses, photographers and optometrists working in the clinics, primary care doctors and optometrists, hospital managers, relevant industry and charity sector professionals, and care commissioners. An approximate target of 40 participants has been set, but the targeting and scale of recruitment will be refined by themes and stakeholders identified in the data.

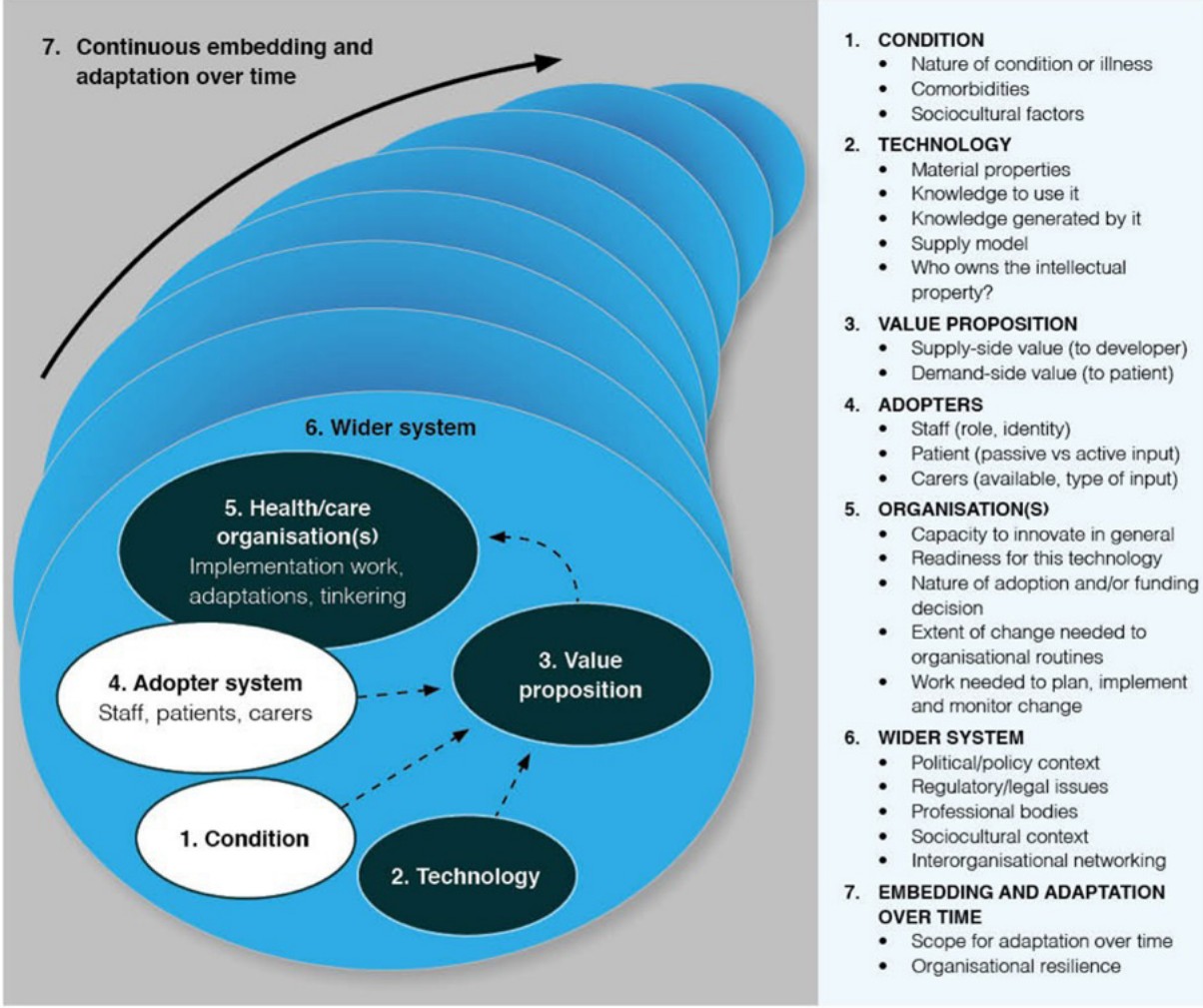

**Figure 3** Schematic of the Non-adoption, Abandonment, Scale-up, Spread and Sustainability (NASSS) framework.[35]

## Data collection

Topic guides have been informed by the aforementioned systematic review and input from clinicians, academics, patients and lay members of the study team (online supplemental file 2). In their initial form, these topic guides focus on:

► Stakeholders' experiences of the present macula service.
► Their perceptions of clinical AI generally and specifically in recommending treatment intervals for nAMD.
► Where this AI-enabled decision tool should be placed within the healthcare pathway and which cases or settings may not be appropriate.
► Factors likely to affect the implementation of this AI-enabled decision tool.
► How relationships between stakeholders and with the AI itself may change or develop.

The topic guides will be used flexibly to guide semi-structured interviews and continuously iterated as data are analysed in parallel. Depending on participant preference interviews may be conducted face-to-face or remotely by telephone or teleconferencing platforms. Interviews will be recorded and transcribed for text-based coding and analysis. Interviews will be delivered by the same researcher (HDJH) throughout, who is an ophthalmologist with lived experience of working in NuTH nAMD clinics and prior training and experience in qualitative research methods. Although patients with prior clinical contact with the interviewer will be ineligible for recruitment and the interviewer will be presented as a researcher rather than a clinician, their status as a healthcare professional may emerge unintentionally and could influence data elicitation.[33] It is also true that some of the healthcare professional interviewees will have worked with the interviewer as a colleague in the past which is likely to influence data. Reflective journaling will be used to transparently report and manage these influences on data collection, analysis and interpretation.[34] Data will also be reviewed throughout the collection period by the wider study team, including lay representatives, to help identify and manage biases in sampling, data collection, analysis and interpretation.

## Data analysis

The same researcher (HDJH) will familiarise themselves with data as it is collected and iteratively place data into a-priori codes and categories derived from a qualitative framework synthesis based on the Non-adoption, bandonment, Scale-up, Spread and Sustainability framework (figure 3).[35] These theoretical and empirical foundations will ensure data are structured in a way that accounts for the policy, organisational and practice level factors that influence implementation.[36] This will help to direct ongoing data collection, but also to rigorously curate data in preparation for thematic analysis.[37] Selection of a theoretical approach for data analysis will be informed by the data and systematic libraries of theoretical approaches used in clinical AI qualitative research

and implementation research generally.[38 39] Consolidated criteria for Reporting Qualitative research (COREQ) reporting guidelines will be used.[40]

## Patient and public involvement

The design of this protocol was informed by patient and carer members of the UK Macular Society and a National Institute for Health and Social Care Research (NIHR) funded lay consumer panel. These groups highlighted opportunities to improve current care by reducing travel requirements for a population commonly affected by other health and social challenges and reducing the congestion of clinics and the time that patients spend there. The project continues to benefit from a study advisory group including an nAMD patient and carer who use the NuTH service and a reference group including four members of the public from across the UK. These individuals have helped design the topic guide and will continue to advise on sampling, data collection, analysis and interpretation.

## Ethics and dissemination

The study has received NHS Research Ethics Committee and UK Health Research Authority approvals (21/NW/0138). Informed consent is planned for interview participants only. Though direct public access to data from the study is not supported, this ethics approval includes a means of sharing anonymised data following requests to the corresponding author and an application process. Dissemination is planned to clinical, academic and commercial stakeholders through peer-reviewed conference presentation and open access journal publication. Public dissemination is planned through NIHR and macular society communication channels and an NIHR funded event in Newcastle upon Tyne for patients and their carers.

**Author affiliations**
[1]Population Health Sciences Institute, University of Newcastle upon Tyne, Newcastle upon Tyne, UK
[2]Newcastle Eye Centre, Newcastle Upon Tyne Hospitals NHS Foundation Trust, Newcastle Upon Tyne, UK
[3]Institute of Ophthalmology, University College London, London, UK
[4]Moorfields Eye Hospital City Road Campus, Moorfields Eye Hospital NHS Foundation Trust, London, UK
[5]Faculty of Business and Law, Northumbria University, Newcastle upon Tyne, UK

**Contributors** HDJH led funding application and drafted the manuscript. All authors contributed to the study design. DT, KBrittain, KBalaskas, SJT, PK and GM reviewed and revised the manuscript.

**Funding** This study was part of a proposal funded by a National Institute for Health Research (NIHR) doctoral fellowship (NIHR301467). The funder had no role in study design, data collection, data analysis, data interpretation or manuscript writing.

**Competing interests** None declared.

**Patient and public involvement** Patients and/or the public were involved in the design, or conduct, or reporting, or dissemination plans of this research. Refer to the Methods section for further details.

**Patient consent for publication** Not applicable.

**Provenance and peer review** Not commissioned; externally peer-reviewed.

**ORCID iD**
Henry David Jeffry Hogg http://orcid.org/0000-0001-8044-7790

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
