## [Reviewer comments · BMJ Open]

ARTICLE DETAILS

TITLE (PROVISIONAL)	Safety and efficacy of an artificial intelligence-enabled decision tool for treatment decisions in neovascular age-related macular degeneration and an exploration of clinical pathway integration and implementation: protocol for a multi-methods validation study
AUTHORS	Hogg, Henry; Brittain, Katie; Teare, Dawn; Talks, S.; Balaskas, Konstantinos; Keane, Pearse; Maniatopoulos, Gregory

VERSION 1 – REVIEW

REVIEWER	Garcin, Thibaud Universite Jean Monnet Saint-Etienne
REVIEW RETURNED	30-Nov-2022

GENERAL COMMENTS	Dear Authors thanks for your paper; This is interesting. Some points need to be explained or precised: p3: please precise the service ; please precise the frequency of the visit ; maybe you should use key word "fusion data ; i.e. clinical + imaging datasets" for the readers and you paper visibility on pubmed or else when published. p4: please add for the findings strength/limitation => results will be "exploratory" due to design of the study. p6: concerning decision treatment and performing injections in UK => different practices throughout different countries (please modulate this in discussion). p6-7: aims are well explained and clear ; but maybe add that the 2° point (qualitative study must/may be the most crucial for future applications ; if the "general" is how change general practices thanks to IA. p8: panel 1 and 2 clear ; but justify why only one single clinician have done review for eligibility criteria ; moreover the same clinician is involved in screening approach with other authors => why the review for eligibility criteria was done by a single person HDJD ? Supplementary file OK // easier and avoiding biais to exclude period since COVID 19 had expanded : good. p9: why did you not mention Phakic status or not , and IOP in data set => not available ? not planned ? / p10: please precise which/nature of IVI (several molecules of antivegf ?) ? p11 : which pilot data was used line 36 ? p12 : outcomes clear. p13 : table and text clear. p14 : methodology OK. p15: justify why sample size was not be powered for secondary outcomes?
---

	p15 end: target and scale of recruitment should be more precised. p16 : detailed interview process is clear and fair / p17: ok for data analysis. p17/18: patients and ethics OK. supplementary files : colocated 25 slice fovea centered OCT available between 2 visits => should be repeated in full text for more precision to the reader and how AI use imaging data as decision tool.
--	---

REVIEWER	Mendes, Luis AIBILI
REVIEW RETURNED	06-Dec-2022

GENERAL COMMENTS	The manuscript presents a clinical protocol for a multi-methods study that aims to establish the safety, efficiency, and operationalization of an AI-based decision tool for the management of nAMD. Using this AI-based approach the authors aim to improve the quality and the capacity of the UK Healthy services. Since the lack of ophthalmologists is not a UK problem, a similar AI-based approach can be applied to other countries. Special care should be considered for the introduction of Ai techniques in critical areas like defense, security, and healthcare. In this particular area is not only important to put an effort in the independent validation of the AI methods but also to know how the models behave in the presence of new data that includes extreme cases. Models' privacy issues should also have into account. The protocol tries to address two problems. The first is related to the AI model to detect disease stability. An AI OCT segmentation model followed by a decision tree will be used and developed/trained to perform the binary classification. Related to the segmentation model is not clear if the final implementation will have a quality check module. Usually, a quality check module is usual when automat methods are introduced in real clinical conditions. Can you detail the approach to be followed to address this problem? Related to the decision tree (a simple and interpretable model) is not clear if the rules will be established based on the pilot study data or are already established and if the pilot data will be only used to establish the cut point/thresholds of the tree. Can you clarify this point? If the structure of the tree is already established, can you provide more details about the criteria/methods and data used to generate the structure? If the tree structure will be established based on the pilot data, how you will ensure that the rules do not change (stability) when training with a different dataset? Although the main idea of this protocol is to study the non-inferiority of AI-based technology in clinical practice, in the real condition the system should detect problematic cases and send them for manual review (like a second opinion). Can you provide some discussion related to this point? Related to the second point, in the process map (figure xx), one output of this study should be identifying the steps that have a major impact on improving the efficiency of the workflow.
---

VERSION 1 – AUTHOR RESPONSE

Reviewer 1's comments

p3: please precise the service ; please precise the frequency of the visit ; maybe you should use key word "fusion data ; i.e. clinical + imaging datasets" for the readers and you paper visibility on pubmed or else when published.

Thank you, we have adapted the methods section of the abstract in line with your helpful comments to provide more detail. We have not specified the range of visit frequencies within the dataset at this stage as whilst we would expect frequencies between 4 and 16 weeks, because of our random sampling approach we do not yet know what the composition of the ultimate dataset will be. We have added an indication of this expected 4 – 16 week range in the 'sampling method' sub-section. Thanks very much indeed too for the key word suggestion.

p4: please add for the findings strength/limitation => results will be "exploratory" due to design of the study.

Thank you, we agree this is an important limitation to state explicitly. Particularly for AI integration where prospective use often reveals problems that retrospective studies fail to uncover. We have added a 5th bullet point to the strengths and limitations section.

p6: concerning decision treatment and performing injections in UK => different practices throughout different countries

Thank you, this is an excellent point. We have added a sentence to the end of the sub-section entitled 'AI-enabled decision tool' to emphasise this limitation and our partial mitigation of it.

p6-7: aims are well explained and clear ; but maybe add that the 2° point (qualitative study must/may be the most crucial for future applications ; if the "general" is how change general practices thanks to IA.

Thank you. We completely agree and welcome the opportunity to emphasise your point. A further sentence has been added to the introduction section, immediately prior to the aim statement.

p8: panel 1 and 2 clear ; but justify why only one single clinician have done review for eligibility criteria ; moreover the same clinician is involved in screening approach with other authors => why the review for eligibility criteria was done by a single person HDJD ?

Thank you for this point, we can see that the clarity of meaning and rationale are could both be improved. The end of the 'sampling method' subsection have been reworked to explain our goal of achieving consensus on the eligibility criteria and their screening application with a multidisciplinary group to satisfy the academic requirements for the method and ensure their meaningful translation into the complex real-world electronic medical records. The rationale for having a single researcher working on data extraction was to draw on their experience with the electronic medical record and clinical and documentation practices at the host institution, but also minimise the influence of inter-researcher variability in data collection.

Supplementary file OK // easier and avoiding biais to exclude period since COVID 19 had expanded : good.

Thank you

p9: why did you not mention Phakic status or not , and IOP in data set => not available ? not planned ?

Thank you for your suggestion. These two variables were not felt to be relevant to the decision process under investigation, with the potential exception of cataract or posterior capsule opacification leading to degradation of OCT quality and potentially affecting AI performance. We chose to address this through post-hoc error analyses to explore failure mechanisms as described in the 'data collection and processing' subsection.

p10: please precise which/nature of IVI (several molecules of antivegf ?) ?

Thank you, this has been added as a third sentence in the 'sampling method' subsection.

p11 : which pilot data was used line 36 ?

Thank you for flagging this clarity issue. We have added some initial detail at this point and signposted to the more thorough description of the pilot dataset later in the manuscript.

P12 : outcomes clear. P13 : table and text clear.

Thank you

p14 : methodology OK.

Thank you

p15: justify why sample size was not be powered for secondary outcomes?

Thank you, a sentence explicitly stating our intention and rationale for powering the sample size to the primary outcome only has been added at the end of the 'justification of stud design and sample size' subsection.

P15 end: target and scale of recruitment should be more precisd.

Thank you, we agree this is desirable. Unfortunately, because of the novelty of the application and research question there are no available data to support an accurate power calculation. We considered proceeding with a power calculation containing assumptions to try and cover this but decided it presented to great a risk of underpowering and an indeterminate study outcome. To mitigate this we have been explicit about the size and nature of the pilot dataset which will inform the power calculation and committed to the other constituents of the power calculation itself. Without delaying publication until the data collection and analysis described have been performed we cannot see other opportunities to meaningfully improve this aspect. We felt the benefit of earlier dissemination to prevent duplication and enable co-ordinations of efforts between researchers outweighed this potential advantage.

P16 : detailed interview process is clear and fair / p17: ok for data analysis.

Thank you

p17/18: patients and ethics OK.

Thank you

supplementary files : colocated 25 slice fovea centered OCT available between 2 visits => should be repeated in full text for more precision to the reader and how AI use imaging data as decision tool.

Thank you for spotting this oversight, we agree it will be helpful to make this aspect more visible. It has been added to the inclusion criteria displayed in panel 2.

Reviewer 2's comments

Related to the segmentation model is not clear if the final implementation will have a quality check module. Usually, a quality check module is usual when automat methods are introduced in real clinical conditions. Can you detail the approach to be followed to address this problem?

Thank you, we agree that this will be a crucial element for any prospective implementation of the proposed AI-enabled decision tool. In this retrospective setting we plan instead to explore all of the tool's false judgements through a post-hoc analysis to attempt to explore the tool's mechanisms of failure and characterise cases for which performance should be specifically reviewed or monitored specifically in future work. A sentence has been added to that effect within the 'data collection and processing' subsection.

Related to the decision tree (a simple and interpretable model) is not clear if the rules will be established based on the pilot study data or are already established and if the pilot data will be only used to establish the cut point/thresholds of the tree. Can you clarify this point?

Thank you, we agree the clarity needs to be improved. Changes have been made within the 'AI-enabled decision tool' subsection to clarify the proposed tissue groups and thresholds planned for the decision tree and the intention to iterate upon this proposed decision tree using the pilot dataset described late in the manuscript.

If the structure of the tree is already established, can you provide more details about the criteria/methods and data used to generate the structure? If the tree structure will be established based on the pilot data, how you will ensure that the rules do not change (stability) when training with a different dataset?

Thank you for identifying this opportunity to improve clarity. We have explicitly mentioned and referenced the 2020 UK consensus report on nAMD treatment protocols on which this initial decision tree is based. The deep learning model within the AI-enabled decision tool will not be altered or trained in any way throughout the study.

Although the main idea of this protocol is to study the non-inferiority of AI-based technology in clinical practice, in the real condition the system should detect problematic cases and send them for manual review (like a second opinion). Can you provide some discussion related to this point?

Thank you, this is extremely important as the tool moves towards implementation. In the aforementioned adjustment to the 'data collection and processing' section we have expanded the purpose of the post-hoc failure mechanism analysis to include the identification of cases which may only be appropriate for clinician judgements. Within the scope of the current study, stakeholder interviews are another important insight into this, which has been signalled more heavily through an adjustment to the topic guide summary in the 'data collection' subsection and the addition of supplementary file 2 – a sample topic guide used for patient participants.

Related to the second point, in the process map (figure xx), one output of this study should be identifying the steps that have a major impact on improving the efficiency of the workflow.

Thank you, we agree and have chosen to explore that from an interpretivist perspective in this study by including stakeholders' perspectives on where best to include the AI-enabled decision support tool within the pathway. We do have ethical approval in place to collect relevant health economic and process time data relating to opportunities identified by participants, but felt that these quantitative assessments could be better targeted following qualitative data analysis.

VERSION 2 – REVIEW

REVIEWER	Garcin, Thibaud Universite Jean Monnet Saint-Etienne
REVIEW RETURNED	28-Dec-2022

GENERAL COMMENTS	Dear Authors, Thanks for this R1 and the complete response to comments. You made and justify all points addressed. R1 is clearer and may be interesting for the reader ship of BMJ open.
---

REVIEWER	Mendes, Luis AIBILI
REVIEW RETURNED	07-Jan-2023

GENERAL COMMENTS	The authors address all the comments. The manuscript is ready to be published. Only one small correction. [page 47] The AUC values vary between 0 and 1 , i.e., are not reported as percentages. (check https://en.wikipedia.org/wiki/Receiver_operating_characteristic for more details)
---

VERSION 2 – AUTHOR RESPONSE

Thank you very much both to the editorial team and the reviewers for the time taken to consider and improve the manuscript. I have submitted two versions of the second revision following this round of comments; changes tracked and clean. This is to address the helpful observation made by reviewer 2:

Only one small correction. [page 47] The AUC values vary between 0 and 1 , i.e., are not reported as percentages. (check https://en.wikipedia.org/wiki/Receiver_operating_characteristic for more details)

I admit, that I struggled to use page 47 as a means of locating the issue as page 47 on the pdf proof appears to me to refer to the end of the 'sampling method' subsection under the 'quantitative methods' section of the 'methods and analysis' section, which makes no reference to AUC. There is however, a single mention of 'area under the ROC curve' on p15 of this proof (5 lines beneath table 1) which makes the error described. Thank you very much for identifying this, I have corrected the 99% to 0.99 as suggested. If there are other instances I have failed to recognise or the reviewer was referring to something else then I apologise for missing it and would be happy to be directed another point in the article.